# LEARNING DYNAMIC ABSTRACT REPRESENTATIONS FOR SAMPLE-EFFICIENT REINFORCEMENT LEARNING

## ABSTRACT

In many real-world problems, the learning agent needs to learn a problem's abstractions and solution simultaneously. However, most such abstractions need to be designed and refined by hand for different problems and domains of application. This paper presents a novel top-down approach for constructing state abstractions while carrying out reinforcement learning. Starting with state variables and a simulator, it presents a novel domain-independent approach for dynamically computing an abstraction based on the dispersion of Q-values in abstract states as the agent continues acting and learning. Extensive empirical evaluation on multiple domains and problems shows that this approach automatically learns abstractions that are finely-tuned to the problem, yield powerful sample efficiency, and result in the RL agent significantly outperforming existing approaches.

## 1    INTRODUCTION

It is well known that *good abstract representations* can play a vital role in improving the scalability and efficiency of reinforcement learning (RL) (Sutton & Barto, 2018; Yu, 2018; Konidaris, 2019). However, it is not very clear how good abstract representations could be efficiently learned without extensive hand-coding. Several authors have investigated methods for aggregating concrete states based on similarities in value functions but this approach can be difficult to scale as the number of concrete states or the transition graph grows.

This paper presents a novel approach for top-down construction and refinement of abstractions for sample efficient reinforcement learning. Rather than aggregating concrete states based on the agent's experience, our approach starts with a default, auto-generated coarse abstraction that collapses the domain of each state variable (e.g., the location of each taxi and each passenger in the classic taxi world) to one or two abstract values. This eliminates the need to consider concrete states individually, although this initial abstraction is likely to be too coarse for most practical problems. The overall algorithm proceeds by interleaving the process of refining this abstraction with learning and evaluation of policies, and results in automatically generated, problem and reward-function specific abstractions that aid learning. This process not only helps in creating a succinct representation of cumulative value functions, but it also makes learning more sample efficient by using the abstraction to locally transfer states' values and cleaving abstract states only when it is observed that an abstract state contains states featuring a large spread in their value functions.

This approach is related to research on abstraction for reinforcement learning and on abstraction refinement for model checking Dams & Grumberg (2018); Clarke et al. (2000) (a detailed survey of related work is presented in the next section). However, unlike existing streams of work, we develop a process that automatically generates conditional abstractions, where the final abstraction on the set of values of a variable can depend on the specific values of other variables. For instance, Fig. 1 displays a taxi world where for different values of the state variables (destination and passengers locations), meaningful conditional abstractions are constructed for the taxi location. A meaningful abstraction provides greater details in the taxi-location variable around the passenger location when the taxi needs to pick up a passenger (Fig. 1 (middle)). When the taxi has the passenger, the abstraction should show greater details around the destination (Fig. 1(right)). Furthermore, our approach goes beyond the concept of counter-example driven abstraction refinement to consider the reward function as well as stochastic dynamics, and it uses measures of dispersion such as the standard deviation of Q-values to drive the refinement process. The main contributions of this paper

are mechanisms for building conditional abstraction trees that help compute and represent such abstractions, and the process of interleaving RL episodes with phases of abstraction and refinement. Although this process could be adapted to numerous RL algorithms, we focus on developing and investigating it with Q-learning in this paper.

The presented approach for dynamic abstractions for RL (DAR+RL) can be thought of as a dynamic abstraction scheme because the refinement is tied to the dispersion of Q-values based on the agent's evolving policy during learning. It provides adjustable degrees of compression (Abel et al., 2016) where the aggressiveness of abstraction can be controlled by tuning the definition of variation in the dispersion of Q-values. Extensive empirical evaluation on multiple domains and problems shows that this approach automatically learns abstract representations that effectively draw out similarities across

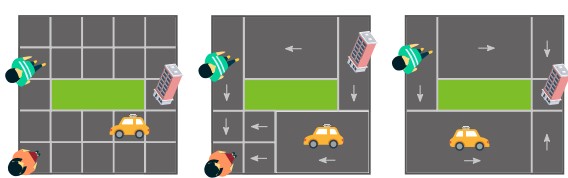

Figure 1: Consider a classic taxi world with two passengers and a building as the drop-off location where the green area is impassable (left). Meaningful conditional abstractions can be constructed, for example, for situations where both passengers are at their pickup locations (middle), or one passenger has already been picked-up (right).

the state space, and yield powerful sample efficiency in learning. Comparative evaluation shows that Q-learning based RL agents enhanced with our approach outperform state-of-the-art RL approaches in both discrete and continuous domains while learning meaningful abstract representations.

The rest of this paper is organized as follows. Sec. 2 summarizes the related work followed by a discussion on the necessary backgrounds in Sec. 3. Sec. 4 presents our dynamic abstraction learning method for sample-efficient RL. The empirical evaluations are demonstrated in Sec. 5 followed by the conclusions in Sec. 6.

## 2    RELATED WORK

**Offline State Abstraction.** Most early studies focus on action-specific (Dietterich, 1999) and option-specific (Jonsson & Barto, 2000) state abstraction. Further, Givan et al. (2003) introduced the notion of state equivalence to possibly reduce the state space size by which two states can be aggregated into one abstract state if applying a mutual action leads to equivalence states with similar rewards. Later on, Ravindran & Barto (2004) relaxed this definition of state equivalence by allowing the actions to be different if there is a valid mapping between them. Offline state abstraction has further been studied for generalization and transfer in RL (Karia & Srivastava, 2022) and planning (Srivastava et al., 2012).

**Graph-Theoretic State Abstraction.** Mannor et al. (2004) developed a graph-theoretic state abstraction approach that utilizes the topological similarities of a state transition graph (STG) to aggregate states in an online manner. Mannor's definition of state abstraction follows Givan's notion of equivalence states except they update the partial STG iteratively to find the abstractions. Another comparable method proposed by Chiu & Soo (2010) carries out spectral graph analysis on STG to decompose the graph into multiple sub-graphs. However, most graph-theoretic analyses on STG, such as computing the eigenvectors in Chiu & Soo's work, can become infeasible for problems with large-scale state space.

**Monte-Carlo Tree Search (MCTS).** MCTS approaches offer viable and tractable algorithms for large state-space Markovian decision problems (Kocsis & Szepesvári, 2006). Jiang et al. (2014) demonstrated that proper abstraction effectively enhances the performance of MCTS algorithms. However, their clustering-based state abstraction approach is limited to the states enumerated by their algorithm within the partially expanded tree, which makes it ineffectual when limited samples are available to the planning/learning agent. Anand et al. (2015) advanced Jiang's method by comprehensively aggregating states and state-action pairs aiming to uncover more symmetries in the domain. Owing to their novel state-action pair abstraction extending Givan and Ravindran's notions of abstractions, Anand et al.'s method results in higher quality policies compared to other approaches based on MCTS. However, their bottom-up abstraction scheme makes their method computation-

ally vulnerable to problems with significantly larger state space size. Moreover, their proposed state abstraction method is limited to the explored states since it applies to the partially expanded tree.

**Counterexample Guided Abstraction Refinement (CEGAR).** CEGAR is a model checking methodology that initially assumes a coarse abstract model and then validates or refines the initial abstraction to eliminate spurious counterexamples to the property that needs to be verified (Clarke et al., 2000). While most work in this direction focuses on deterministic systems, research on the topic also considers the problem of defining the appropriate notion for a "counterexample" in stochastic settings. E.g., Chadha & Viswanathan (2010) propose that a counterexample can be considered as a small MDP that violates the desired property. However, searching for such counterexamples can be difficult in the RL setting where the transition function of the MDP is not available. Seipp & Helmert (2018) developed algorithms for planning in deterministic environments that invoke the CEGAR loop iteratively on the same original task to obtain more efficient abstraction refinement. These methods do not consider the problem of building abstractions for stochastic planning or reinforcement learning.

## 3 BACKGROUND

Markov decision Processes (MDPs) (Bellman, 1957; Puterman, 2014) are defined as a tuple $\langle \mathcal{S}, \mathcal{A}, \mathcal{T}, \mathcal{R}, \gamma \rangle$, where $\mathcal{S}$ and $\mathcal{A}$ denote the state and action spaces respectively. Generally, a concrete state $s \in \mathcal{S}$ can be defined as a set of $n$ state variables such that $\mathcal{V} = \{v_i | i = 1, ..., n\}$. In this paper, we focus on problems where the state is defined using a set of variables. An extension to partially observable settings where the agent receives an image of the state is a promising direction for future work. $\mathcal{T} : \mathcal{S} \times \mathcal{A} \times \mathcal{S} \rightarrow [0, 1]$ is a transition probability function, $\mathcal{R} : \mathcal{S} \times \mathcal{A} \rightarrow \mathbb{R}$ is a reward function, and $\gamma$ is the discount factor. The unknown policy $\pi$ is the solution to an MDP, denoted as $\pi : \mathcal{S} \rightarrow \mathcal{A}$. We consider the RL settings where an agent needs to interact with an environment that can be modeled as an MDP with unknown $\mathcal{T}$. The objective is to learn an optimal policy for this MDP.

When the size of the space state increases significantly, most of the RL algorithms fail to solve the given MDP due to the *curse of dimensionality*. Abstraction is a dimension reduction mechanism by which the original problem representation maps to a new reduced problem representation (Giunchiglia & Walsh, 1992). We adopt the general definition of state abstraction proposed by Li et al. (2006).

**Definition 1** *Let $M = \langle \mathcal{S}, \mathcal{A}, \mathcal{T}, \mathcal{R}, \gamma \rangle$ be the ground MDP from which the abstract MDP $\bar{M} = \langle \bar{\mathcal{S}}, \mathcal{A}, \bar{\mathcal{T}}, \bar{\mathcal{R}}, \gamma \rangle$ can be derived via a state abstraction function $\phi : \mathcal{S} \rightarrow \bar{\mathcal{S}}$, where the abstract state mapped to concrete state $s$ is denoted as $\phi(s) \in \bar{\mathcal{S}}$ and $\phi^{-1}(\bar{s})$ is the set of concrete states associated to $\bar{s} \in \bar{\mathcal{S}}$. Further, a weighting function over concrete states is denoted as $w(s)$ with $s \in \mathcal{S}$ s.t. for each $\bar{s} \in \bar{\mathcal{S}}$, $\sum_{s \in \phi^{-1}(\bar{s})} w(s) = 1$, where $w(s) \in [0, 1]$. Accordingly, the abstract transition probability function $\bar{\mathcal{T}}$ and reward function $\bar{\mathcal{R}}$ are defined as follows:*

$$\bar{\mathcal{R}}(\bar{s}, a) = \sum_{s \in \phi^{-1}(\bar{s})} w(s)\mathcal{R}(s, a), \qquad \bar{\mathcal{T}}(\bar{s}, a, \bar{s}') = \sum_{s \in \phi^{-1}(\bar{s})} \sum_{s' \in \phi^{-1}(\bar{s})} w(s)\mathcal{T}(s, a, s').$$

When it comes to the decision-making in an abstract MDP, all concrete states associated with an abstract state $\bar{s} \in \bar{\mathcal{S}}$ are perceived identically. Accordingly, the relation between abstract policy $\pi : \bar{\mathcal{S}} \rightarrow \mathcal{A}$ and the concrete policy $\pi : \mathcal{S} \rightarrow \mathcal{A}$ can be defined as $\pi(s) = \bar{\pi}(\phi(s))$ for all $s \in \mathcal{S}$. Further, the value functions for an abstract MDP are denoted as $V^{\bar{\pi}}(\bar{S})$, $V^*(\bar{S})$, $Q^{\bar{\pi}}(\bar{S}, a)$, and $Q^*(\bar{S}, a)$. For more on RL and value functions see Sutton & Barto (2018), for MDPs, see (Bellman, 1957; Puterman, 2014), and for more on the notion of abstraction, refer Giunchiglia & Walsh (1992); Li et al. (2006).

## 4 OUR APPROACH

### 4.1 OVERVIEW

Starting with state variables and a simulator, we develop a domain-independent approach for dynamically computing an abstraction based on the dispersion of Q-values in abstract states. The idea of dynamic abstraction is to learn a problem's solution and abstractions simultaneously. We propose a

top-down abstraction refinement mechanism by which the learning agent effectively refines an initial coarse abstraction through acting and learning. We illustrate this mechanism with an example.

**Example 1** *Consider a 4x4 Wumpus world consisting of a pit at (2,2) and a goal at (4,4). In this domain, every movement has a reward -1. Reaching the goal results in a positive reward of 10 and the agents receive a negative reward -10 for falling into the pit. The goal and the pit are the terminal states of the domain. The agent's actions include moving to non-diagonal adjacent cells at each time step s.t.* $\mathcal{A} = \{up, down, left, right\}$.

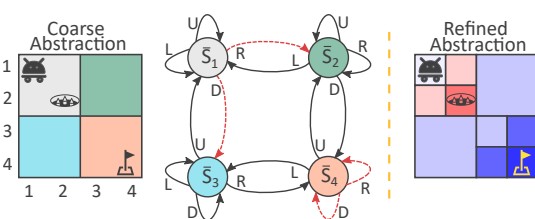

Figure 2: An example of dynamic heterogeneous abstraction refinement for a Wumpus world.

Considering Example 1, Fig. 2 (left) shows a potential initial coarse abstraction in which the domain of each state variable (here horizontal and vertical locations) is split into two abstract values and $\bar{S}_1$ and $\bar{S}_4$ contain the pitfall and goal location respectively. As a result, when learning, the agent will observe a high standard deviation on the values of $Q(\bar{S}_1, right), Q(\bar{S}_1, down), Q(\bar{S}_4, right)$, and $Q(\bar{S}_4, down)$ because of the presence of terminal states with large negative or positive rewards. Guided by this dispersion of Q-values, the initial coarse abstraction should be refined to resolve the observed variations. Fig. 2 (right) exemplifies an effective abstraction refinement for Example 1 demonstrated as a heatmap of Q-values. Notice that the desired abstraction is a heterogeneous abstraction on the domains of state variable values where the abstraction on a variable depends on the value of the other variables: let $x$ and $y$ be the horizontal and vertical locations of the agent in Example 1 respectively and their domain be $\{1, 2, 3, 4\}$. When $y > 2$, the domain of $x$ (originally $\{1, 2, 3, 4\}$) is abstracted into sets $\{1, 2\}, \{3\}$, and $\{4\}$, but when $y \leq 2$, the domain of $x$ is abstracted into sets $\{1\}, \{2\}$, and $\{3, 4\}$.

The next section (Sec. 4.2) presents our novel approach for automatically computing such abstractions while carrying out RL.

## 4.2 CONDITIONAL ABSTRACTION TREES

The value of a state variable $v_i$ inherently falls within a known range. Partitioning these ranges is one way to construct state abstractions. However, in practice, the abstraction of one state variable is conditioned on a specific range of any other state variables. Accordingly, we need to maintain and update such conditional abstractions via structures that we call Conditional Abstraction Trees (CATs) while constructing the state abstractions.

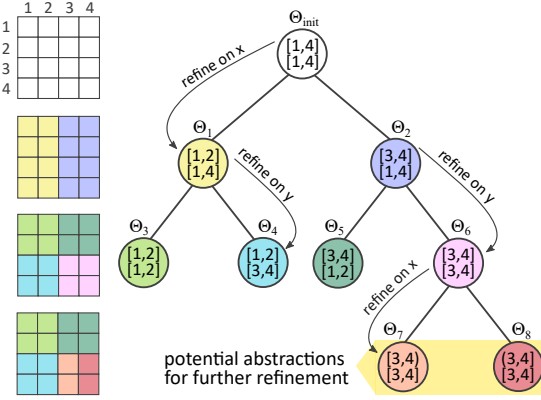

Figure 3: This figure illustrates a Conditional Abstraction Tree (CAT) for Example 1. Ranges written inside the nodes represent $\theta_i \in \Theta$. Each node represents a conditional abstraction.

Fig. 3 (right) exemplifies a partially expanded CAT for the problem in Example 1. This problem can be represented by two state variables, i.e. agent's horizontal and vertical location denoted as $x$ and $y$ respectively. The tree's root node contains the global ranges (The first range refers to the horizontal location $x$ and the second range refers to the vertical location $y$) for both of these state variables representing an initial coarse abstraction (in white). The annotations visualize how this initial abstraction can be further refined w.r.t a state variable resulting in new conditional abstractions (symmetric annotations are not shown for the sake of readability). The refinement procedure of the Wumpus world associated to each level of the tree is also displayed in Fig. 3 (left).

Given the set of state variables $\mathcal{V}$, we define an abstract state using the set of partitions, one for each variable $v_i$, where each partition $\theta_i$ is an interval of the form $[l_i, h_i]$. Thus, a 2-D abstract state for Example 1 could be defined by $\theta_1 = [1, 4]$ and $\theta_2 = [1, 2]$. An abstraction is defined as $\Theta = \{\theta_i | i \in [1, n]\}$, where $n = |\mathcal{V}|$. In fact, CAT is a hierarchical abstraction tree starting with an initial abstraction $\Theta_{init}$ that represents the original range for each state variable $v_i \in s$ s.t. $\Theta_{init} = \{\theta_i | i \in [1, n]$ and $l_i = v_i^{min}$ and $h_i = v_i^{max}\}$, where $v_i^{min}$ and $v_i^{max}$ denote the lower and upper bounds on the range of $v_i$ respectively. In Example 1, there are two state variables so the initial abstraction is $\Theta_{init} = \{[1, 4], [1, 4]\}$. The initial abstraction also induces the starting coarse abstraction since the range for each state variable suggests that all values for all state variables is compressed into one abstract state.

This initial coarse abstraction induced by the initial abstraction $\Theta_{init}$ needs to be further refined so that the learning agent can improve its performance through a more fined representation of the problem. We define a refinement function $\delta(\Theta, i, f)$ that splits the range of partition $\theta_i \in \Theta$ of state variable $v_i$ into $f$ equal ranges resulting in $f$ new abstractions. Now, we formally define the refinement function $\delta(\Theta, i, f)$.

**Definition 2** *Let $\Theta = \langle \theta_1, \ldots, \theta_n \rangle$ be an abstract state for a domain with variables $v_1, \ldots v_n$. We define the f-split refinement of $\Theta$ w.r.t. variable $i$ as $\delta(\Theta, i, f) = \{\Theta^1, \ldots, \Theta^f\}$ where all $\Theta^j$'s are the same as $\Theta$ on $\theta_k$ for $k \neq i$. $\theta_i = [l, h]$ is partitioned with $f$ new boundaries at least $\|\theta\|/f$ values apart: $l, l_1, l_2, \ldots, l_f, h$ where $l_x = l + x \times \lfloor [(h - l)/f] \rfloor$.*

Next, we need to define the relation between two given abstractions in the form of $\Theta$ in order to determine if one is obtained by refining the other one.

**Definition 3** *Let $\Psi$ be the set containing all possible abstractions. Given $\Theta_a, \Theta_b \in \Psi$, we say $\Theta_b$ is obtained by refining $\Theta_a$, denoted as $\Theta_b \triangleright \Theta_a$, iff $(\forall i \in [1, n])(\theta_i^b \subseteq \theta_i^a)$. Moreover, $\Theta_b \triangleright \Theta_a \equiv \Theta_a \triangleleft \Theta_b$. Although this definition determines an ancestral relation between $\Theta_a$ and $\Theta_b$, we need to know the factor $f$ by which $\Theta_a$ has been refined to determine if $\Theta_b$ is the direct result of refining $\Theta_a$. We say $\Theta_b$ is obtained directly by refining $\Theta_a$, denoted as $\Theta_b \unrhd \Theta_a$, iff $\exists i (\theta_i^b \subset \theta_i^a)$, $(\forall k_{\neq i} \in [1, n])(\theta_k^b = \theta_k^a)$ and $|\theta_i^b| \times f = |\theta_i^a|$.*

With these definitions in hand, we can now formally define CAT as an undirected tree to construct and maintain the hierarchy of the conditional partitions. Conditional Abstraction Tree (CAT), denoted as $\xi$, represents a tree structure specifying the topology between conditional abstractions in the form of $\Theta$.

**Definition 4** *A conditional abstraction tree (CAT) is defined as $\xi = \{N, E\}$, where $N$ is the set of nodes and $E$ is the set of edges. Each node in $N$ corresponds to an abstraction $\Theta$, s.t. $N = \{\Theta_m | m \in [1, n_\xi]\}$, where $n_\xi$ is the cardinality of CAT, and $\Theta_{root} = \Theta_{init}$, where $\Theta_{root}$ is the root node of the tree. Every parent $\Theta_p$ and child $\Theta_c$ nodes in $\xi$ are connected via an edge $e_p^c$ s.t. $e_p^c \implies \Theta_c \unrhd \Theta_p$. Additionally, $L_\xi = \{\Theta_m | (\forall k \in [1, n_\xi])(\Theta_m \ntrianglerighteq \Theta_k)\}$ is defined as the set of leaf nodes. $L_\xi$ represents the set of abstract states in $\xi$.*

Given CAT $\xi$ and the value of a concrete state $s$, the mapping $\phi(s) : \mathcal{S} \to \bar{\mathcal{S}}$ can be done via a level-order tree search starting from $\Theta_{root}$. The corresponding abstract state $\bar{s}$ is in the node $\Theta_{found}$ iff $\forall i \in [1, n]$ $v_i \in \theta_i^{found}$ (inclusion condition) and $\Theta_{found}$ is a leaf node, i.e. $\Theta_{found} \in L$. Alg. 1 computes the $\phi : \mathcal{S} \to \bar{\mathcal{S}}$ mapping for a given concrete state $s$ under CAT $\xi$, starting from CAT's root node $\Theta_{root}$. `FindAbstract`$(\xi, \Theta_{start}, s)$ starts the level-order search from $\Theta_{start}$ and it always finds the corresponding abstract state when $\Theta_{start} = \Theta_{root}$. This algorithm checks the inclusion condition first for $\Theta_{start}$ (Line 1 in Alg. 1). If $\Theta_{Start}$ is not a leaf node,

---

**Algorithm 1:** State Abstraction

**FindAbstract** (CAT $\xi$, $\Theta_{start}$, $s$):

1: **if** $(\forall v_i \in s)(v_i \in \theta_i^{start})$ **then**
2:     **if** $\Theta_{start} \in L_\xi$ **then**
3:         **return** $\Theta_{start}$
4:     **else**
5:         $children \leftarrow Children(\Theta_{start})$
6:         **for** $\Theta_{child} \in children$ **do**
7:             **if** $(\forall v_i \in s)(v_i \in \theta_i^{child})$ **then**
8:                 **FindAbstract** $(\xi, \Theta_{child}, s)$

---

the algorithm checks the inclusion condition for children of $\Theta_{start}$ (Line 7 in Alg. 1) and if a child satisfies the condition, `FindAbstract` gets invoked recursively (Line 8 in Alg. 1).

Any state abstraction under a given CAT $\xi$ induces an abstract representation of the underlying concrete MDP $M$. Thus, an MDP $M$ can have two abstract representations $\bar{M}_a$ and $\bar{M}_b$ under two CATs $\xi_a$ and $\xi_b$ respectively. We define a relational operation to decide which abstract MDP is finer.

**Definition 5** *Given MDPs $\bar{M}_a$ and $\bar{M}_b$ abstracted under $\xi_a$ and $\xi_b$, we say $\bar{M}_a$ is strictly finer than $\bar{M}_b$, denoted as $\bar{m}_a \succ \bar{m}_b$, iff $\forall \Theta^a \in L_{\xi_a} \ \exists \Theta^b \in L_{\xi_b} \ (\Theta^a \trianglerighteq \Theta^b)$. We also say $\bar{M}_a$ is finer than $\bar{M}_b$, denoted as $\bar{M}_a \succeq \bar{M}_a$, iff $\forall \Theta^a \in L_{\xi_a} \ \exists \Theta^b \in L_{\xi_b} \ (\Theta^a \trianglerighteq \Theta^b \vee \Theta^a = \Theta^b)$.*

### 4.3 LEARNING DYNAMIC ABSTRACTIONS

Definition 4 formalizes the abstraction tree by which the mapping $\phi(s) : \mathcal{S} \to \bar{\mathcal{S}}$ can be performed using a level-order search (see Alg. 1), while Definition 2 explains how a node of a CAT can be refined with respect to a state variable $v_i$ through the refinement function $\delta(\Theta, i, f)$. However, our objective is to interleave RL episodes with phases of abstraction refinement leading to an enhanced abstract policy $\bar{\pi}$ for a given concrete MDP $M$. We need to develop a mechanism that 1) observes the dispersion of Q-values while the RL agent is acting and learning through the abstract MDP $\bar{M}$, and 2) refines unstable abstract states w.r.t a state variable.

Therefore, our approach, Dynamic Abstractions for RL (DAR+RL), consists of three phases: 1) the RL agent performs Q-learning over the abstract state space $\bar{\mathcal{S}}$ defined by leaves of the current CAT and learns an abstract policy $\bar{\pi}$; 2) the RL agent continues interacting with the environment via the abstract policy $\bar{\pi}$ and DAR+RL evaluates the computed abstraction by observing the dispersion of Q-values; and 3) DAR+RL refines the current abstraction by finding unstable abstract states in $\xi$. One needs to blame a state variable $v_i$ for each unstable state since the refinement can be conducted w.r.t to one state variable as defined in Definition 2.

Let $\beta(M, \xi, \bar{\pi})$ denote the evaluation function which is simply an RL routine where the learning agent interacts with an MDP $M$ through a fixed policy $\bar{\pi}$ under the abstraction computed by $\xi$ for one single episode; Throughout one episode of evaluation, the observed dispersion of Q-values is defined as $D = \{d_m | m \in [1, n_{step}]\}$. The observed dispersion $D$ is the set of observed $Q^{\bar{\pi}}(\bar{S}, a)$ values for one episode (up to $n_{step}$ steps) of evaluation.

---

**Algorithm 2:** Learning Dynamic Abstractions

**Input**: $M, f$
**Output**: $M, \xi, \bar{\pi}$
1: initialize $\Theta_{init}$, $\xi$, and $\bar{Q}$
2: **for** $episode = 1, n_{epi}$ **do**
3:    $s \leftarrow$ reset()
4:   **for** $steps$ in $episode$ **do**
5:      $\bar{s} \leftarrow$ FindAbstract($\xi, \Theta_{init}, s$)
6:      $a \leftarrow \bar{\pi}(\bar{s})$
7:      $s', \bar{r}, done \leftarrow$ step(extend($a$))
8:      $\bar{s}' \leftarrow$ FindAbstract($\xi, \Theta_{init}, s'$)
9:      $\bar{\pi} \leftarrow$ train$^{\bar{\pi}}(\bar{s}, \bar{s}', a, \bar{r})$
10:     $s, \bar{s} \leftarrow s', \bar{s}'$
11:   **if** $M$ needs refinement **then**
12:     **for** $e = 1, n_{eval}$ **do**
13:       $\Gamma$.append(evaluate($M, \xi, \bar{\pi}$))
14:     $unstable \leftarrow$ UnstableState($\Gamma$)
15:     **for** each $\Theta$ in $unstable$ **do**
16:       $i \leftarrow$ UnstableVar($\Gamma, \Theta$)
17:       $nodes \leftarrow$ refine($\Theta, i, f$)
18:       $\xi \leftarrow$ UpdateTree($\xi, \Theta, nodes$)
19: **return** $M, \xi, \bar{\pi}$

---

To obtain a better exploration over the abstract states, the evaluation function $\beta(M, \xi, \bar{\pi})$ needs to be executed for $n_{eval} > 1$ episodes. That being said, $\Gamma$ denotes all observed dispersion obtained by executing the evaluation function for $n_{eval}$ episodes, where $\Gamma = \{D_m | m \in [1, n_{eval}]\}$. Let UnstableState($\Gamma$) denote a function that finds the set of unstable states in the form of $\Theta$ based on the dispersion of Q-values in $\Gamma$. Besides, UnstableVar($\Gamma, \Theta$) denotes a function that finds the accountable state variable for each unstable state, given the dispersion log $\Gamma$. Altogether, within the DAR+RL learning, evaluation, and refinement phases, the learning agent learns the solution to the MDP $M$ while learning the dynamic abstractions.

### 4.4 DAR+RL ALGORITHM

Alg. 2 illustrates the procedure by which the agent learns an MDP's solution and abstractions simultaneously through learning, evaluation, and refinement phases explained in Sec. 4.3. First, the initial coarse abstraction needs to be automatically constructed through initializing $\Theta_{init}$, based on the known ranges for each state variable $v_i$, and constructing a CAT $\xi$ with only the root node (Line 1 in Alg. 2). The initial $\xi$ induces an abstract MDP $\bar{M}$ for the given MDP $M$.

Then, the learning phase of DAR+RL starts by employing the Q-learning routine (Lines 2 to 10 in Alg. 2). In other words, throughout the learning phase (lines 2 to 2) Alg. 2 implement the vanilla Q-learning over abstract states computed from the CAT. In this phase, induced by the computed state abstraction, extended actions (taking a concrete action repeatedly until the agent reaches a new abstract state, blockage, or a terminal concrete state) are applied to the environment instead of the concrete actions (Line 7 in Alg. 2). As the result, the agent enhances the abstract policy $\bar{\pi}$ to learn the solution to the abstract MDP $\bar{M}$.

Since the initial abstraction is likely to be too coarse, DAR+RL checks the refinement condition (Line 11 in Alg. 2) at the end of each learning episode to initiate an evaluation phase followed by a refinement phase. We set DAR+RL to check the recent success rate of the RL agent every $n_{check}$ episodes where the refinement condition evaluate to true if the success rate is below some threshold. The choice of the refinement condition introduces a trade-off. On one hand, we want to obtain a near optimal abstraction that enables the agent to learn the solution effectively. On the other hand, the abstract policy $\bar{\pi}$ should be trained enough to be used in the evaluation phase for refinement purposes. When the refinement condition is true, the algorithm runs the evaluation function $\beta$ (a standard Q-learning routine with a fixed policy) for $n_{eval}$ episodes (Line 13 in Alg. 2). During the evaluation phase, it is likely to encounter an abstract state $\bar{s}$ multiple times. Since the policy is fixed in this phase, comparing different Q-values for the same pair of state-action $(\bar{s}, \bar{\pi}(a))$ exposes potential inconsistencies in the abstract state $\bar{s}$. To capture such inconsistencies, for all observed abstract states, DAR+RL logs different computed Q-values and store them in $\Gamma$. Then, for each abstract state $\bar{s}$ in $\Gamma$, DAR+RL calculates the normalized standard deviation of all logged $Q(\bar{s}, \bar{\pi}(a))$ (Line 14 in Alg 2). Considering these calculated values, `UnstabelState`$(\Gamma)$ finds the unstable states (abstract states with high variations) using clustering techniques. Finally, the unstable states are refined and the abstraction tree is updated accordingly in Lines 15 to 18.

## 5 EMPIRICAL EVALUATIONS

To assess the performance of DAR+RL, we conducted empirical analysis on three discrete domains: Office World adapted from Icarte et al. (2018), Wumpus World derived from Russell & Norvig (2020), Taxi World adapted from the OpenAI Gym environment Taxi-v3 (https://www.gymlibrary.ml/environments/toy_text/taxi/) and introduced by Dietterich (2000), and one continuous domain: Water World based on Karpathy (2015); Icarte et al. (2018). All of the domains used are stochastic continuous/discrete problems with varying dimensionality (from 2 to 14). All the details regarding the domains and task descriptions are included in the supplementary document. We aim to investigate the following:

- Does DAR+RL improve the sample efficiency of vanilla Q-learning without any expert knowledge?

- Is DAR+RL scalable to high-dimensional tasks?

- Does DAR+RL recognize similar abstractions for similar sub-problems in a larger task?

For the comparative study, we selected the following baselines: (1) Option-critic Bacon et al. (2017), (2) JIRP Xu et al. (2020), (3) tabular Q-learning Watkins & Dayan (1992), (4) DQN Mnih et al. (2013), (5) A2C Mnih et al. (2016), and (5) PPO Schulman et al. (2017). Option-critic is a Hierarchical RL (HRL) approach that discovers options autonomously while learning option policies simultaneously. JIRP automatically infers reward machines and policies for RL. We chose these state-of-the-art methods as baselines as they do not require expert knowledge as input. We also compared with deep RL methods: DQN, A2C, and PPO. The details of parameters and hyperparameters are included in the supplementary document.

For each domain, we executed 10 independent runs for each algorithm and report the mean success rates averaged over last 100 training episodes along with the standard deviations. We also report normalized cumulative reward for each domain and method obtained by evaluating the agent on 10 simulation runs, after stopping training at intervals of 10 episodes. We use implementation of DQN, A2C, and PPO from the Stable-Baselines3 by Raffin et al. (2019). Our code is included in the supplementary material. We now discuss our results and analysis in detail below.

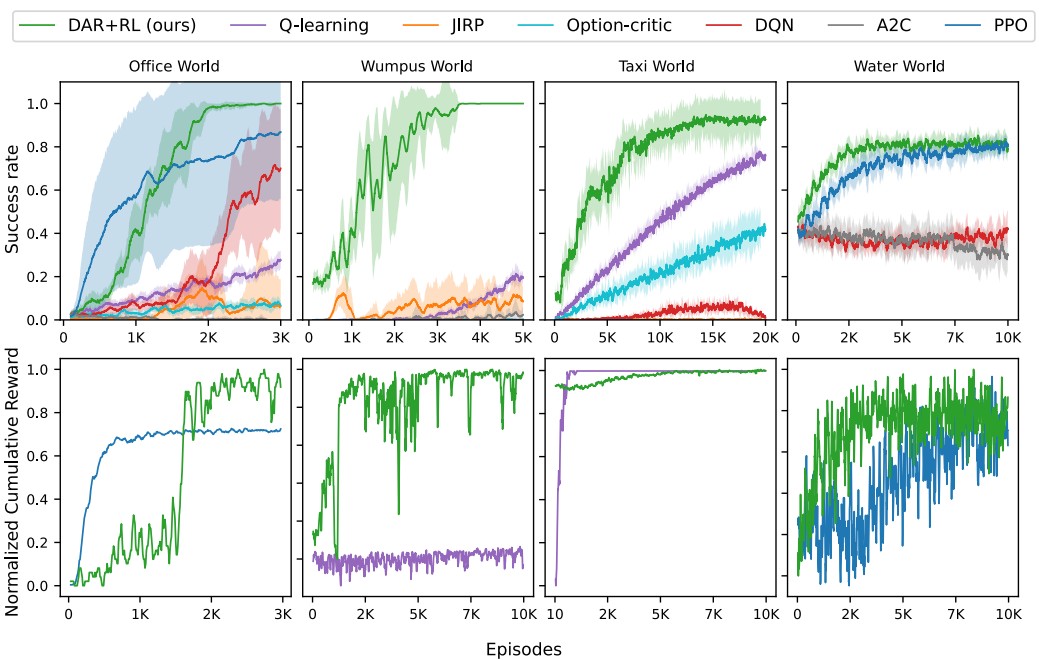

Figure 4: (Top) Success rates (mean and standard deviation) for 10 independent runs averaged over last 100 training episodes for all the methods, and (Bottom) normalized cumulative reward for 10 simulation runs obtained every 10 training episodes for DAR+RL (ours) and the second-best performing baseline for Office World (36x36), Wumpus World (64x64), Taxi World (30x30), and Water World (300x300) domains.

## 5.1 RESULTS

Fig. 4 (Top) shows comparison among success rates achieved by all the methods on all the domains. In Office world, DAR+RL outperforms all the baselines and almost converges to a success rate of 1 in around 2000 episodes, whereas, PPO reaches an approximate success rate of 0.8 in 2500 episodes and has a high standard deviation. DQN reaches a success rate of only 0.65 within 3000 episodes and rest of the baselines struggle to learn and are stuck below 0.2 success rate. In Wumpus world, DAR+RL converges to success rate of 1 within 4000 episodes and significantly outperform all the baselines which are stuck below a success rate of only 0.2. In Taxi world, DAR+RL achieves a success rate of almost 1.0 within 12000 episodes of training, while Q-learning and Option-critic perform better than other baselines achieving approximate success rates of 0.75 and 0.4 respectively within 20000 episodes. Even in the continuous Water world domain, DAR+RL learns slightly faster than PPO while all other baselines perform poorly and are stuck below a success rate of 0.4. We performed further evaluation on DAR+RL and the second-best performing baseline on each domain as shown in the Fig. 4 (Bottom) by evaluating the policies learned by the agent. In Office and Water worlds, DAR+RL outperforms PPO, and in Wumpus world, DAR+RL gains significantly higher cumulative reward than Q-learning.

## 5.2 ANALYSIS

**Sample efficiency in the absence of input expert knowledge.** The results presented in Section 5.1 demonstrate that DAR+RL's performance is superior to all baselines in both discrete and continuous domains. This is categorically the effect of the learned conditional abstractions by DAR+RL made available to the vanilla Q-learning algorithm. This effect can be perceived from two perspectives: 1) the meaningful conditional abstractions that are automatically constructed by DAR+RL spotlight the most informative aspects of the state space, leading to more sample-efficient learning; and 2) the Q-learning agent benefits from significantly higher levels of exploration over state and action spaces due to the nature of abstraction. This intense exploration can cause more penalization of the agent

at the early stages of learning (see cumulative rewards of DAR+RL in taxi and office world) but eventually leads to faster learning and superior performance reflected in the success rate.

**Scalability to high-dimensional tasks.** RL algorithms that learn policy $\pi$ from a concrete MDP $M$ suffer from the curse of dimensionality as the size of the state space increases. This explains why most of the baselines fail to learn the Wumpus world, as a basic domain, when the size of the grid increases drastically, as shown in Fig. 4. In contrast, the top-down abstraction refinement scheme of DAR+RL scales effectively to problems with relatively larger state space. As a result, the abstract representations learned by DAR+RL empowered vanilla Q-Learning algorithm to learn those problems relatively fast and efficiently. We conducted further experiments on scalability and computational complexity of DAR+RL and baselines and the results are presented in the supplementary document.

**Abstractions learned in similar sub-problems.** One important property of DAR+RL's framework is to construct identical abstractions across the state space for similar sub-problems. This capability of DAR+RL is critically beneficial in large problems where options can be generalized across identically constructed abstractions. Fig. 5 demonstrates two constructed conditional abstractions by DAR+RL for an 8×8 taxi world. In Fig. 5 (middle), the passenger is located at top-left and the destination is located at the bottom-left of the map. Besides, in Fig. 5 (right), the passenger is in the taxi and the destination is

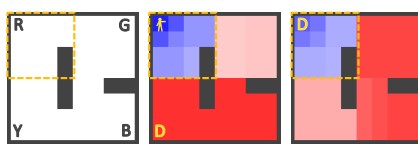

Figure 5: Drawing out similarities across state space of a 8×8 taxi world via DAR+RL's automatic abstraction.

located at the top-left. In both cases, the agent should reach the top-left cell of the map which implies a similarity. DAR+RL discovered this similarity automatically as seen from the generated identical abstractions (highlighted area) for both the cases.

## 6    CONCLUSION

We presented a novel approach (DAR+RL) for simultaneously learning dynamic abstract representations along with the solution to problems formulated as an MDP. The overall algorithm of DAR+RL proceeds by interleaving the process of refining a coarse initial abstraction with learning and evaluation of policies for the underlying RL agent (Q-learning). Besides, we introduced conditional abstraction trees to compute and represent such refined abstractions throughout the DAR+RL procedure. Extensive empirical evaluation on multiple domains of problems demonstrated that DAR+RL effectively enables vanilla Q-learning algorithm to learn the solution to large discrete and continuous problems, with dynamic representations, where state-of-the-art RL algorithms are outperformed. This superior performance of vanilla Q-learning compared to algorithms with complex neural-network-based architecture such as PPO and A2C is due to DAR+RL's scalable abstraction construction scheme that effectively draws out similarities across the state space and yields powerful sample efficiency in learning. Future work will consider automatic discovery of generalizable options utilizing the constructed conditional abstract representations by DAR+RL.

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
