# OpenReview forum: "LEARNING DYNAMIC ABSTRACT REPRESENTATIONS FOR SAMPLE-EFFICIENT REINFORCEMENT LEARNING"
_ICLR.cc/2023/Conference — Submitted to ICLR 2023_

### Official Review · Reviewer_3y7Q · 2022-10-15

**Confidence:** 4
**Correctness:** 2
**Technical Novelty And Significance:** 2
**Empirical Novelty And Significance:** 2
**Recommendation:** 5

**Clarity, Quality, Novelty And Reproducibility:**

CLARITY/QUALITY:
The paper lacks clarity in the description of the approach.

A few key strange elements:
-The paper is unclear about how the interleave between Q-learning and refinement should happen. This is also not clear in the description of the algorithms. For instance "Then, the learning phase of DAR+RL starts by employing the Q-learning routine (Lines 2 to 10 in Alg. 2)" does not seem to actually describe what is going on in the algorithm from lines 2 to 10 where we can't see any information about Q-values.
-It is also unclear why this whole procedure would be faster than learning directly without state abstractions. It seems that learning the Q-values for every refinement of the state space is a costly approach.
-The formalization of the approach (mainly starting from the top of page 5) is not clearly written. The formalization is interleaved with examples that only help moderately to understand what is actually done.

Some examples of details that are wrong or not clear:
- (section 3) The paper writes "$\mathcal T :S \times A \rightarrow [0,1]$ is a transition probability function". Mapping a state-action pair into $[0,1]$ does not look like a well-defined transition function.
- (algorithm 1) I can't understand what is supposed to mean by "1: if (∀vi ∈ s)(vi ∈ θstart) then"

NOVELTY:
-The paper does not compare with any of the methods that actually aim at doing state abstraction (it compares with Option-critic , JIRP, tabular Q-learning, DQN, A2C and PPO.
-The authors discuss the related work, however they do not highlight why their method is different/better. For instance, the authors mention in the end of the related work section that "These methods do not consider the problem of building abstractions for stochastic planning or reinforcement learning." It seems that this paper has some of the same limitations (deterministic environment) but nothing is said about it.

REPRODUCIBILITY:
Source code does not seem to be provided. Based on the paper, I don't think reproducibility is feasible.


**Strength And Weaknesses:**

Strengths:
- The paper makes use of simple examples to motivate the approach.

I have two main concerns.
- My first concern is related to the writing of the paper and the lack of clarity with respect to some of the key components.
- Second, I have doubts about the actual procedure that the paper describes.

**Summary Of The Paper:**

The paper aims at iteratively computing an abstraction based on the dispersion of Q-values for states that have been aggregated together.  The paper describes the algorithm with a formalization and some pseudo-codes. The algorithm is tested on different discrete and deterministic grid-world environements.

**Summary Of The Review:**

I have two main concerns.
- The writing of the paper and the lack of clarity with respect to some of the key components.
- I have doubts about the actual procedure that the paper describes (the formalization and the pseudo code do not provide enough details).

---

> ### Author Response · Authors · 2022-11-15
> **Response to Reviewer 3y7Q**
>
> Thank you for your feedback. We have incorporated your suggestions for clarity in the algorithm. We would like to clarify that our source code is included with the main paper. We now address your concerns below.
>
>
> ### Notation clarity:
> Thank you for pointing out the typo in the definition of the transition function. We fixed this issue in the revised version of the paper. We would like to clarify your concern with line 1 in Algorithm 1 as well. Basically, Algorithm 1 searches for a mapping from a given concrete state to a leaf node in CAT (an abstract state). Line 1 checks if the values of the given concrete state falls within the specified ranges in $\theta_i^{start}$.
>
>
> ### Learning abstract Q-values:
> We want to clarify that DAR+RL does not learn the Q-values for every refinement of the state space. Instead, it learns the Q-values only for the set of abstract states constructed by the conditional abstract tree. The worst case complexity of the learning and evaluation phases of DAR+RL is similar to that of the underlying RL algorithm that it is used (Q-learning in this case). The refinement phase consists of a CAT search for an unstable state with a time complexity of $O(n\log n)$ and a split operation which is linear in the number of state variables. Practically speaking, we found DAR+RL to be surprisingly efficient in terms of runtime. We conducted further experiments on different problem sizes for Office World and found that although Q-learning completed before our approach for small problems, DAR+RL is significantly faster than Q-learning when the problem size increases even when the time for abstraction refinement is taken into account. These results are added to the Sec. 2 in the supplementary document. Furthermore, the plots in Fig. 4 also include the episodes that were spent on intermediate evaluation of standard deviation of the abstract Q-function. Thus, the comparison clearly shows that DAR+RL leads to massive improvements in sample efficiency of vanilla Q-learning compared to state-of-the-art methods. The reason for this performance boost is that, in practice, DAR+RL performs significantly fewer computations than it would require to solve the underlying MDP due to the abstraction that it builds on the fly. Although the abstract MDP becomes finer after each refinement phase, the state space size of this abstract MDP is still significantly smaller than the concrete MDP.
>
>
> ### Details of DAR+RL’s components:
> Thank you for pointing out the need for clarity on UnstableState function and a few other components of the paper. We noticed a few writing issues and have included more detail in both the paper and the supplementary document (Sec. 4). To summarize, UnstableSate function takes all samples obtained during the evaluation phase and calculates the normalized standard deviation of Q-values for each abstract state. Then, it clusters out the top k unstable states using off-the-shelf clustering algorithms. More details about the methods we developed to blame a state variable for an unstable state are also included in the supplementary document in Sec. 4.
>
>
> ### Selection of baselines:
> We conducted an extensive literature review on state abstraction and reported them under multiple categories in the Related Work section. In fact, most of these works have structural limitations, such as requiring hand-coded inputs or offline operations, that make them inapplicable to the relevant RL problems without the additional inputs. On the other hand, we selected different state-of-the-art deep RL methods as the baselines since multiple layers in their neural network architectures progressively construct state abstractions [15]. We also chose state-of-the-art symbolic RL (JIRP) and hierarchical RL (Option-critic) methods for comparison as they automatically learn different abstract representations such as reward machines and options to expedite learning without requiring any human-engineered inputs, similar to our method. We welcome suggestions for other specific baselines.
>
>
> ### Reproducibility:
> We respectfully believe that DAR+RL is reproducible since we had provided the source code in the supplementary materials.
>
> #### references
>
> [15] Mnih et al. Human-level control through deep reinforcement learning. nature, 2015.

---

> > ### Comment · Reviewer_3y7Q · 2022-11-15
> > **Thanks for the clarifications/modifications. I increase my score from 3 to 5 (reject to borderline)**
> >
> > Thanks for the clarifications and for the modifications of the paper.
> >
> > Reproducibility: Thanks for pointing out that the source code is also shared. As the availability of the source code might be an important point I would suggest that this information is also given somewhere in the main paper (not only in the appendix).
> >
> > Given all that I'm willing to increase my score from reject (3) to borderline (5). Even though some of the comments I've made in my review were taken into account, I think that the overall writing and level of details are still a bit weak for a strong scientific contribution. In addition, other reviews also pointed valid concerns that might not have received full attention (discussion with other reviewers is also needed here), which is why I do not increase my score higher than borderline for now.

---

> > > ### Author Response · Authors · 2022-11-16
> > > **Thank you for your response!**
> > >
> > > Thank you for letting us know that we have addressed your major concerns and for updating your score. We look forward to your comments after the discussion with other reviewers.

---

### Official Review · Reviewer_4Xkd · 2022-10-25

**Confidence:** 3
**Correctness:** 2
**Technical Novelty And Significance:** 3
**Empirical Novelty And Significance:** Not applicable
**Recommendation:** 5

**Clarity, Quality, Novelty And Reproducibility:**

This paper was very clearly written.  It is novel to the best of my knowledge.  However, I have concerns about the quality; see above.

**Strength And Weaknesses:**

Minor nitpick: in the definition of \mathcal V, doesn’t the notation i \in [1, n] imply that i takes a continuous range of real values in that range?  Wouldn’t it be more correct to write {1, 2, …, n}?  (Please ignore this if I’m incorrect.)

Definition 5: I think there may be a few errors here.  After the first “denoted as”, there are two a’s (shouldn’t there be an “a” and a “b”?).  Also, are l_a and l_b defined?  I’m aware of the definition above where they are a scalar lower bound of an interval for a theta_i, but that does not seem their usage in this definition.

Strengths:
- This paper was exceptionally clear and well-written.
- The included code is a nice addition; it appears to be well-organized.

Weaknesses:
1. Minor weakness: I believe the paper overstates the general utility of this approach.  For example, consider a task in a 3-D world where the observation is pixel input.  Clearly, the authors’ approach will not be helpful in this case, since DAR+RL will attempt to divide the world up based on uninformative ranges of individual pixels.  This concern is not limited to pixel input.  This is not to say that this approach is not a valuable contribution, but I wonder whether many real-world problems with high-dimensional state spaces can be made more tractable by this approach.  Weakness 4 below is closely related to this. I am concerned that the authors are overselling their approach and sweeping its limitations under the rug.  Instead, I’d advise frankly discussing these limitations, discussing future work which might allow one to overcome them, and evaluating on more diverse environments (again, see weakness 4).
    - Of course, DAR+RL will be more effective if the state/observation is designed to be more suitable to it (e.g., use x/y coordinates rather than pixel inputs), but hand-designing the state/observation like this defeats the intended purpose of DAR+RL.
2. Minor weakness: Given the well-known high variance between RL runs, 10 trials is not ideal to demonstrate the claims being made.  However, I am not overly concerned about this in this case, given the plots (which seem to show clear trends) and the more pressing concerns below.
3. Major weakness: Not all hyperparameters are included (particularly the architectures’ hyperparameters, but also details about the optimizers, etc.).  This is closely related to weakness 5.
4. Major weakness: All domains are some variety of 2-D gridworlds.  This limits the convincingness of the empirical work and is closely related to weaknesses 1 and 5.  It also makes the paper’s claims about “high-dimensional tasks” unconvincing, since these environments all seem to have 1-2 dozen state dimensions at most, while a large proportion of modern RL focuses on tasks with hundreds or thousands of state or observation dimensions.  One fix for a future submission would be to include higher-dimensional (non-gridworld) tasks such as mujoco domains to the experiments.
5. Major weakness: Some important details about the baselines are unclear to me.  Particularly, the details of the architectures used are missing.  I’m not sure any of the baselines are fair because:
    - The tabular Q-learning approach will, of course, not perform well given so many possible states.
    - The following bullet points assume that the other methods all took real-valued x and y inputs to the neural nets (the same type of inputs that DAR+RL received), and used some standard deep architecture from stablebaselines3.  I believe all baselines other than tabular Q-learning fall into this category.  If this is not the case, the situation could be better or worse, depending on the details:
        - The deep RL approaches will not perform well without hyperparameter tuning (which seems to have been done for DAR+RL, but not the baselines).
        - Worse, these architectures are designed for much more complex environments, and so will perform relatively poorly in terms of data efficiency compared to a “quasi-tabular” approach such as DAR+RL, since their networks are excessively large and overparameterized for these type of problems.  Better baselines might be linear function approximators (see the Sutton and Barto book, 2nd edition, for a good list of effective ones) or perhaps very small neural networks, since these approaches can be expected to learn these these simple environments, with their relatively-small-number-of “continuous” state dimensions (most are not truly continuous, but the agents observe continuous values), more efficiently than large deep nets.
6. Minor weakness: I do not believe the details of the UnstableState function (line 14 of algorithm 2) used in the experiments were given.  These details, including relevant hyperparameters, should be provided.


**Summary Of The Paper:**

The authors propose a method which divides the state space into discrete “abstractions”, based on variance of the state action values.  They compare their method to several baselines.


**Summary Of The Review:**

This paper is very well-written.  However, despite all the notation, its contribution is entirely empirical, and I have concerns about the quality of the empirical work.

---

> ### Author Response · Authors · 2022-11-15
> **Response to Reviewer 4Xkd**
>
> Thank you for all your feedback and suggestions. We have incorporated all your suggestions about notational clarity. Thank you for providing those.
>
> We address your other comments below.
>
> ### Details of DAR+RL’s components:
> Thank you for pointing out the need for clarity on $\texttt{UnstableState}()$ function and a few other components of the paper. We noticed a few writing issues and have included more detail in both the paper and the supplementary document (Sec. 4).
>
> ### Notations:
> Thank you for mentioning these. We have resolved the notional issues you pointed out. Regarding Definition 5, $l_a$ and $l_b$ are the sets of leaf nodes for CATs $\xi_a$ and $\xi_b$ respectively. We revised these notations to fix the duplication. Now, the leaf nodes of $\xi_a$ and $\xi_b$ are denoted as $L_{\xi_a}$ and $L_{\xi_b}$ respectively.
>
> ### Pixel input:
> Kindly see the common note on image-based state representations. We agree with you that this work may not be well suited for image-based representations without further optimizations. We would like to point out that there is a vast body of contemporary, state-of-the-art work focused on scaling up RL to tasks whose states cannot be easily expressed as images or robot configuration states [1-14 in common note]. Using Mujoco (which is a *deterministic* simulator) or image-based representations is not necessary to address the challenges with scalability of RL on non-image-based problems discussed in this work. The vast body of related work cited in the common note above constitutes ample evidence that the problems we address are indeed challenging, relevant to practical scenarios (e.g., mars rover missions, taxi management and many others) and of interest to the research community. We have clarified this in the paper in Sec. 2.
>
> ### Test problems being too simple for deep RL methods:
> Your hypothesis that the deep RL approaches “will perform poorly in terms of data efficiency because they are designed for much more complex environments” was intriguing. We conducted experiments with smaller (less “complex”) versions of our test problems to test this hypothesis. Deep RL approaches should have performed even worse if the hypothesis was true. However, the performance of PPO improved as the size of the OfficeWorld problems was reduced! We include these plots in the supplementary document Sec. 1. This indicates that (a) deep RL methods do better on smaller problem instances that are less “complex” and are unable to handle increasing complexity as problems sizes increase to the levels presented in the paper and (b) methods designed for image-based RL do not directly scale in other RL problems, which is also confirmed by the existence of the vast body of work on such problems (cited above).
>
> ### Dimensionality of test problems:
>  We have selected the test problems in order to introduce different levels of complexity to our method and baselines. Although some of these problems look similar, they represent distinct challenges and dimensions ranging from 2 to 14. We added more details about the domains in the supplementary document.
>
> ### Parameters and hyperparameters:
> We included detailed information about the hyper parameters used for both the baselines and DAR+RL in the supplementary document in Sec. 3. We used standard architectures for A2C, PPO, DQN from StableBaselines3 (https://github.com/DLR-RM/stable-baselines3) and Option-Critic (https://github.com/lweitkamp/option-critic-pytorch). We use the open-source code available for the state-of-the-art baseline JIRP (https://github.com/logic-and-learning/AdvisoRL).
>
> ### Test problems:
> Please see the common response on the selection of test problems.
>
> ### Selection of baselines:
> We selected different state-of-the-art deep RL methods as the baselines since multiple layers in their neural network architectures progressively construct state abstractions [15]. We also chose state-of-the-art symbolic RL (JIRP) and hierarchical RL (Option-critic) methods for comparison as they automatically learn different abstract representations such as reward machines and options to expedite learning without requiring any human-engineered inputs, similar to our method. We welcome suggestions for other specific baselines.
>
> ### Limitations:
> As the first paper developing and evaluating the concept of conditional abstraction trees, this paper evaluates the algorithm when used with vanilla Q-learning. Although the results outperform state-of-the-art RL algorithms in the test problems, further research is needed to enable using DAR+RL with other RL algorithms as well. This is a good direction for future work on the topic.
>
> ### Empirical results being limited to metrics:
> An extended analysis of the empirical results was presented in Sec 5.2 in the submitted version. This section analyzes sample efficiency, scalability, as well as the ability of this approach to discover similar abstractions for similar subproblems across the state space.

---

> > ### Comment · Reviewer_4Xkd · 2022-11-18
> > **Thank you for the response**
> >
> > Thank you for your detailed response.
> >
> > ## Pixel Input:
> >
> > > "It is not true that a problem has to have pixel-based representations to be challenging or relevant to practical deployments of RL." -Authors' response
> >
> > > "**This concern is not limited to pixel input.** This is not to say that this approach is not a valuable contribution, but I wonder whether many real-world problems with high-dimensional state spaces can be made more tractable by this approach." -Original review above, emphasis mine.
> >
> > Again, my concern is not specific to pixel-based inputs.  My concern is that “All domains are some variety of 2-D gridworlds”.  Reviewer 6Ztx had an identical concern: “grid-based environments with additional variables”.  The authors’ response in “RL beyond image-based representations” isn’t really addressing our concerns: We are not asserting that “a problem has to have pixel-based representations to be challenging or relevant to practical deployments of RL”. Therefore, we do not need to be convinced that non-image-based problems are of interest to the community, and an argument that non-image-based problems are of interest does not address our concerns.
> >
> > As I stated in my original review, I think the authors need other kinds of environments to support their claims.  If the authors do not want to extend their work beyond grid-based environments, and think that their approach is only applicable to grid-based environments, then another approach for a future submission could be to clearly limit the claims of the paper (everywhere from the abstract to the introduction to the conclusion) to grid-based environments.
> >
> > ## Test problems being too simple for deep RL methods:
> >
> > Thank you for your hard work on section 1 of the supplementary material.  However, those results are not surprising to me, and do not contradict the quote in my original review: “will perform relatively poorly in terms of data efficiency compared to a “quasi-tabular” approach such as DAR+RL”.  This statement is about data efficiency, and is entirely consistent with all the algorithms (Q-learning, deep networks, and DAR+RL) getting better returns on simpler environments when data is limited.
> >
> > ## Dimensionality of test problems:
> > See above.
> >
> > ## Parameters and hyperparameters:
> > Thank you for including these, I apologize if I missed them in the original submission.  However, this does not address my concerns about hyperparameter tuning for the DAR+RL but not the baselines.
> >
> > ## Summary:
> > The paper has been improved, and I will update my score to a 5, but I still cannot recommend acceptance (see above).

---

> > > ### Author Response · Authors · 2022-11-19
> > > **Thank you for your response!**
> > >
> > > Thank you for your response and for updating your score. We address your other comments below.
> > >
> > > ### Pixel Input:
> > > 1. We have already extended our approach beyond grid-based test problems. As reported in Empirical Evaluations, we conducted experiments on Water World, a domain with **continuous state space**, and demonstrated that DAR+RL performs superior to baselines such as PPO and DQN.
> > > 2. The references [1-14] in the common note are not just general examples of non-image-based problems that are of interest to the community, they are also specific examples of contemporary and state-of-the-art RL works that used **exactly** the same test problems as we did in our empirical evaluations (TaxiWorld [10,13,14], Office World: [1,3,4,5,6,7,8,9,10], Water World: [1,2,4,5], and Wumpus World: [11,12]).
> > >
> > > ### Parameters and hyperparameters:
> > > 1. As we have mentioned in the supplementary document, we did **not** conduct any extensive hyperparameter or parameter tunning for DAR+RL. One important advantage of DAR+RL over Deep-RL baselines is that DAR+RL has few parameters and performs robustly regardless of the value of its parameters as long as they are not set to drastically large or small values within their ranges.
> > > 2. We have conducted extensive hyperparameter tunning for all baselines including deep RL baselines. In fact, a few of the hyperparameters (such as maximum episode length and exploration rate) are set in favor of baselines. We have done hyperparameter tuning for the baselines with respect to the values reported in the contemporary literature and our empirical evaluations.
> > > 3. We explored various neural network architectures for the deep RL baselines and chose the one that led to their best performance (e.g. 2 layers with 64 neurons per layer for PPO). We replicated the scalability study with an alternative neural network architecture (fewer neurons per hidden layer) and included the results in the supplementary document (Fig. 2).
> > >
> > > ### Test problems being too simple for deep RL methods:
> > > This hypothesis that the test problems are too simple for deep RL methods is originally related to your concerns about "overparameterized and excessively large" networks of the deep RL baselines in this paper. First, we would like to clarify that the largest network used in the deep RL baselines is PPO's NN with 2 hidden layers and 64 neurons per hidden layer which is not excessively large for the test problems such as Water World with 14 state variables. Secondly, we conducted an additional set of experiments for PPO with a relatively smaller network (2 hidden layers with only 16 neurons per hidden layer) to replicate the scalability study in Sec. 1 of the supplementary document. We found that smaller network architecture does not improve the performance of PPO which indicates that the original network architectures of deep RL baselines that we used for our empirical evaluations are fairly suitable for the given test problem. Thirdly, the scalability study (mentioned in Sec. 1 of the supplementary document) demonstrates that neither changing the size of the test problems (in terms of state space) nor changing the size of the neural networks does not improve the performance of the deep RL baselines compared to DAR+RL.

---

> > > > ### Comment · Reviewer_4Xkd · 2022-11-21
> > > > **Thank you for your response!**
> > > >
> > > > > Water World
> > > >
> > > > While I agree that the continuous states make this domain a bit less "Gridworldy" than the other domains, it still closely resembles a Gridworld with continuous states.  So this does not alleviate my concern about domain diversity.  As implied above, if this were a less empirical paper, the lack of domain diversity might be okay, but since the contribution is entirely empirical, this is a concern for me.
> > > >
> > > > > hyperparameters
> > > >
> > > > I see; thank you for pointing this out, this alleviates my concern about hyperparameter tuning.
> > > >
> > > > However, given the lack of domain diversity (and my accompanying concern that DAR+RL might be somewhat uniquely suited to these "grid-based environments with additional variables"), the lack of challenging domains with hundreds or thousands of state dimensions, and the fact that only 10 trials were used for each plot, I still feel that the empirical contribution is a bit weak (and the empirical contribution is the entire contribution).  Therefore, I'm unwilling to raise my score any higher.  I do think this is an interesting line of work and encourage the authors to work on addressing the reviewers' concerns and then resubmitting.

---

> > > > > ### Author Response · Authors · 2022-11-23
> > > > > **Thank you for your response!**
> > > > >
> > > > >
> > > > > Thank you very much for your comments and clarification.
> > > > >
> > > > > > While I agree that the continuous states make this domain a bit less “Gridworldy” than the other domains, it still closely resembles a Gridworld with continuous states.
> > > > >
> > > > > The theoretical development of this paper and its empirical evaluation focuses on multi-variable domains (Sec. 3). This is a well established class of non-image-based problems in RL research and constitutes a topic of active research interest as noted in several contemporary papers [1-14]. The problem instances used in this paper are significantly larger than those used in papers addressing such problems.
> > > > >
> > > > > > However, given the lack of domain diversity (and my accompanying concern that DAR+RL might be somewhat uniquely suited to these “grid-based environments with additional variables”), the lack of challenging domains with hundreds or thousands of state dimensions, and the fact that only 10 trials were used for each plot, I still feel that the empirical contribution is a bit weak (and the empirical contribution is the entire contribution).
> > > > >
> > > > > The reviewer may have overlooked this, but our main contributions are (a) a new, formally well-defined class of abstractions (conditional abstraction trees or CATs) and (b) a principled approach for learning CATs on-the-fly for RL on domains with multi-valued variables. This is discussed formally in Sections 4.2 and 4.3 and informally in paras 2-4 of the Introduction. Our evaluation is empirical as is the case for most state-of-the-art research on the topic.
> > > > > Since the reviewer is convinced that non-image-based problems are of interest to the community, we would like to mention that the technical approach presented in this paper outperforms existing approaches on popular, well-established benchmarks for such problems. We welcome suggestions and/or pointers to existing RL work that addresses problems with hundreds or thousands of state dimensions in non-image-based domains.

---

### Official Review · Reviewer_6Ztx · 2022-10-31

**Confidence:** 4
**Correctness:** 2
**Technical Novelty And Significance:** 2
**Empirical Novelty And Significance:** 1
**Recommendation:** 3

**Clarity, Quality, Novelty And Reproducibility:**

Clarity: The paper is generally clear.

Originality: The general idea (iterative course-to-fine refinement of the state space) is an old one. The main contribution of the paper is the specific approach taken but I find that some critical elements of the approach is not fully clear. These include how to determine whether M needs refinement (line 11 in Algorithm 2), how to determine whether a state is unstable (the authors note that this is "based on the dispersion of Q-values"), and how to determine which variables are accountable for the unstable state. These issues are very briefly discussed at the end of Section 4 but are not fully specified.

Quality: Experimental evaluation is on a rather narrow range of environments and does not go into much depth.

The experimental results do little more than present performance metrics. Furthermore, they treat the proposed algorithm as a monolithic structure, without any exploration of its individual components.

For example, it would be useful to see more detail on how to determine whether a state is unstable. The authors note that this is "based on the dispersion of Q-values" and very briefly discuss a possible approach at the end of Section 4. It would be useful to study this aspect of the algorithm in isolation, testing a variety of candidate mechanisms.

While experimental results are presented in four different domains, three of these domains are structurally similar to each other (grid-based environments with additional variables). In order to evaluate how widely useful the proposed approach might be, it would be useful to see experiments in a broad range of domains.

**Strength And Weaknesses:**

Strengths: Abstraction in reinforcement learning is an important area of research; the general approach explored here (iterative course-to-fine refinement of the state space) is plausible.

Weaknesses:

-- Some critical parts of the algorithms are not fully specified. These include how to determine whether M needs refinement (line 11 in Algorithm 2), how to determine whether a state is unstable (the authors note that this is "based on the dispersion of Q-values"), and how to determine which variables are accountable for the unstable state. These issues are very briefly discussed at the end of Section 4 but are not fully specified.

-- The experimental results do little more than present performance metrics. Furthermore, they treat the proposed algorithm as a monolithic structure, without any exploration of its individual components, of the impact of specific algorithmic choices (versus plausible alternatives), and the sensitivity to any parameters.

-- While experimental results are presented in four different domains, three of these domains are very similar to each other (grid-based environments with additional variables).

-- The computational complexity of the approach has not been explored.


**Summary Of The Paper:**

The authors propose an approach to abstraction in reinforcement learning. The proposed approach starts with a coarse state abstraction and iteratively refines it. The basis for the refinement is the dispersion in the Q-values as the agent continues to learn. The approach is empirically evaluated in three grid-based domains (Office World, Taxi World, Wumpus World) and one domain with continuous state variables (Water World).


**Summary Of The Review:**

The paper presents a plausible approach to an important problem in reinforcement learning. However, important algorithmic components are not fully specified. Furthermore, the experimental evaluation is narrow in scope, both in the types of domains tested and in the analyses conducted in each domain. As a result, the strengths and weaknesses of the algorithm are not adequately understood.

---

> ### Author Response · Authors · 2022-11-15
> **Response to Reviewer 6Ztx**
>
> Thank you for your detailed comments and notes about the value of abstractions in RL. We address your other comments below.
>
>
> ### Abstraction refinement in RL:
> You are right that the notion of abstraction refinement in itself is not new. This is noted in Sec. 2. One of our primary contributions is the conditional abstraction tree framework, which enables efficient learning of heterogeneous abstractions on-the-fly unlike the standard approach of defining a single abstraction layer over the given input representation. This allows the computed abstractions to vary depending on the latent subproblem being solved by the agent. E.g., when the taxi needs to pick up a passenger, we want greater detail in the taxi-location variable when it is close to the passenger location. When the taxi has the passenger, we want the abstraction to have greater detail around the destination (Fig. 1). To our knowledge, this is the first approach to represent and learn such abstractions for RL on-the-fly with no hand-coded inputs aside from the standard RL formulation. Fig. 1 and Sec. 1 have been updated to clarify this.
>
>
> ### Details of DAR+RL’s components:
> Thank you for pointing out the need for clarity on $\texttt{UnstableState}()$ function and a few other components of the paper. We noticed a few writing issues and have included more detail in both the paper and the supplementary document (Sec. 4).  To summarize, $\texttt{UnstableState}()$ takes all samples obtained during the evaluation phase and calculates the normalized standard deviation of Q-values for each abstract state. Then, it clusters out the top k unstable states using off-the-shelf clustering algorithms. More details about the methods we developed to blame a state variable are also included in the supplementary.
>
>
> ### Key strengths and weaknesses:
>
> #### Strengths
>
> 1. Computational complexity: The worst case complexity of the learning and evaluation phases of DAR+RL is similar to that of the underlying RL algorithm that it is used (Q-learning in this case). The refinement phase consists of a CAT search for an unstable state with a time complexity of O(nlogn) and a split operation which is linear in the number of state variables. Practically speaking, we found DAR+RL to be surprisingly efficient in terms of runtime. We conducted further experiments on different problem sizes for Office World and found that although Q-learning completed before our approach for small problems, DAR+RL is significantly faster than Q-learning when the problem size increases even when the time for abstraction refinement is taken into account. These results are added to the Sec. 2 in the supplementary document. Furthermore, the plots in Fig. 4 also include the episodes that were spent on intermediate evaluation of standard deviation of the abstract Q-function. Thus, the comparison clearly shows that DAR+RL leads to massive improvements in sample efficiency of vanilla Q-learning compared to state-of-the-art methods. The reason for this performance boost is that, in practice, DAR+RL performs significantly fewer computations than it would require to solve the underlying MDP due to the abstraction that it builds on the fly. Although the abstract MDP becomes finer after each refinement phase, the state space size of this abstract MDP is still significantly smaller than the concrete MDP.
>
>
> 2. Limited parameters and tuning requirements: DAR+RL has very few parameters or hyperparameters. The parameters are $n_{check}$ and the threshold value for the refinement condition, the cap $k$ for the maximum number of unstable states that can be refined in each refinement phase, and $n_{eval}$ for the duration of the evaluation phase. The same parameter values were used across all our experiments except for cap k which was set proportionally to the size of the problem. In contrast we had to conduct significant hyper-parameter exploration for the baselines because the default settings led to insignificant learning.
>
>
> #### Weaknesses:
> As the first paper developing and evaluating the concept of heterogeneous abstractions using conditional abstraction trees, this paper evaluates the algorithm when used with vanilla Q-learning. Although the results outperform state-of-the-art RL algorithms in the test problems, further research is needed to enable using DAR+RL with other RL algorithms as well. This is a good direction for future work on the topic.
>
>
> ### Test problems:
> The test problems used in our experiments are well established as challenging problems for state-of-the-art RL (TaxiWorld [10,13,14], Office World: [1,3,4,5,6,7,8,9,10], Water World: [1,2,4,5], and Wumpus World: [11,12]) algorithms. The WaterWorld problem features several continuous state variables. The versions of Office World, Wumpus World, and Taxi World used in this paper are significantly more challenging than those used in prior work in terms of the ranges of state variables. Kindly see the common note on test problem selection.

---

### Author Response · Authors · 2022-11-15
**Response to All Reviewers**

We thank all the reviewers for detailed comments and suggestions. The main concerns of the reviewers have been addressed in the revised version and the supplementary document. We discuss the common comments here and the specific comments in the response to each reviewer. We wish to clarify a few major points of misunderstanding in some of the reviews.

### RL beyond image-based representations:
The significant body of current RL literature on non-image-based problems (including the test problems used in this paper) [1-14] indicates that this is a challenging and relevant area of investigation. It is not true that a problem has to have pixel-based representations to be challenging or relevant to practical deployments of RL. For instance, the state of a real-world taxi management service would require extensive human effort to be expressed as an image. While there has been a lot of progress on pixel-based state spaces, our extensive empirical analysis clearly shows that performance on non-image based state representations is currently limited. This work extends the scalability of RL on such problems.

### Test problems:
The test problems used in this paper include well-established, stochastic problems with discrete as well as continuous variables with varying dimensionalities (2 to 14). These problems are drawn from contemporary, state-of-the-art publications and they represent problems that are widely considered to be challenging by the research community [1-14].

References:

[1] Icarte et al. Using reward machines for high-level task specification and decomposition in reinforcement learning. In ICML, 2018.

[2] Karpathy, 2015. REINFORCEjs: WaterWorld demo.

[3] Xu et al. Joint inference of reward machines and policies for reinforcement learning. In ICAPS, 2020.

[4] Icarte et al. Reward machines: Exploiting reward function structure in reinforcement learning. In JAIR, 2022.

[5] Camacho et al. LTL and Beyond: Formal Languages for Reward Function Specification in Reinforcement Learning. In IJCAI, 2019.

[6] Illanes et al. Symbolic plans as high-level instructions for reinforcement learning. In ICAPS, 2020.

[7] Furelos-Blanco et al. Induction of subgoal automata for reinforcement learning. In AAAI, 2020.

[8] Dann et al.. Multi-Agent Intention Progression with Reward Machines. In IJCAI, 2022.

[9] Jin et al. Creativity of ai: Automatic symbolic option discovery for facilitating deep reinforcement learning. In AAAI, 2022.

[10] Kokel et al. Reprel: Integrating relational planning and reinforcement learning for effective abstraction. In ICAPS, 2021.

[11] Abel et al. Value preserving state-action abstractions. In ICAIS, 2020.

[12] Barreto et al. Fast reinforcement learning with generalized policy updates. In NAS, 2020.

[13] Bai et al. Efficient reinforcement learning with hierarchies of machines by leveraging internal transitions. In IJCAI, 2017.

[14] Lyu et al.. SDRL: interpretable and data-efficient deep reinforcement learning leveraging symbolic planning. In AAAI, 2019.

---

### Decision · Program_Chairs · 2023-01-20

**Decision:**

Reject

**Justification For Why Not Higher Score:**

None of the reviewers was enthusiastic, the paper didn't seem conceptually hugely novel, and there were concerns about validation.

**Justification For Why Not Lower Score:**

N/A

**Metareview: Summary, Strengths And Weaknesses:**

This paper describes a method for constructing abstraction for reinforcement-learning problems online, during learning, by pursuing a coarse-to-fine search for good partitions of state variables, which yield efficient and accurate learning.

The reviewers found the paper to provide interesting ideas, but had a number of concerns along various lines, including:
- relative similarity of test domains
- a lack of clear articulation of what properties of a problem would make it amenable to these techniques

Fundamentally, the method seems similar in spirit to a line of old work in RL, starting with Moore (PartiGame, NeurIPS 1993), and running through Will Uther's PhD thesis on tree-based approximations (CMU, 2002; or see "TreeBased Discretization for Continuous State Space RL, Uther and Veloso, AAAI98) and surely beyond.  It would be important to compare and contrast your method to these strategies.